# Improvement in Gait and Participation in a Child with Angelman Syndrome after Translingual Neurostimulation Associated with Goal-Oriented Therapy: A Case Report

**DOI:** 10.3390/children9050719

**Published:** 2022-05-13

**Authors:** Delphine Gaudin-Drouelle, Laetitia Houx, Mathieu Lempereur, Sylvain Brochard, Christelle Pons

**Affiliations:** 1LaTim UMR 1101, Team Beachild, INSERM, 29200 Brest, France; laetitia.houx@chu-brest.fr (L.H.); mathieu.lempereur@univ-brest.fr (M.L.); sylvain.brochard@chu-brest.fr (S.B.); christelle.ponsbecmeur@ildys.org (C.P.); 2Department of Pediatric Rehabilitation, Brest University Hospital, 29200 Brest, France; 3Department of Pediatric Rehabilitation, Ildys Fondation, 29200 Brest, France; 4Faculty of Medicine, University of Western Brittany UBO, 29200 Brest, France

**Keywords:** Angelman, translingual neurostimulation, gait analysis, rehabilitation, case report

## Abstract

Angelman syndrome is a genetic neurobehavioral syndrome characterized by motor and cognitive developmental delay, with a severe reduction in activity and participation. Treatments are limited and the effects of rehabilitation have not been studied. We report on the progress made by a 7-year-old boy with Angelman syndrome following an innovative synergic intervention involving translingual neurostimulation (TLNS) and goal-oriented rehabilitation to improve gait. The parents were interviewed regarding the child’s abilities and participation level and three-dimensional gait analysis was performed before and after the 4-week intervention (five days per week, 4 h per day) and 6 months later. Spatiotemporal and kinematic gait variables improved considerably at 4 weeks, with a reduction in lower limb agonist-antagonist co-contractions, and a large increase in walking distance (from 500 m to 2 km). The child’s engagement and ability to perform activities of daily living improved, as well as several functions not targeted by the intervention. Six months after cessation of the intervention, improvements were partially sustained. The rapid and considerable improvement in motor ability was likely due to potentiation of the rehabilitation by the TLNS. Further studies are required to understand the mechanisms underlying this effect and to determine if it is generalizable to other children with similar disorders.

## 1. Introduction

Angelman syndrome (AS) is a rare neurobehavioral, genetic syndrome (estimated prevalence of 1/12,000–1/25,000 births) characterized by severe balance and coordination disorders, delayed overall development, no speech, epilepsy, sleep disturbance, a happy demeanor with frequent laughter, hyperactivity and stereotyped sensory-seeking behaviors.

Angelman syndrome is usually suspected before two years of age when signs of general developmental delay and characteristic behavior (joyful demeanor, hyperexcitability) become apparent and when severe epilepsy with electro-encephalographic patterns occurs. The diagnosis is confirmed by genetic analysis with a characteristic 15q anomaly (deletion, duplication, mutation or imprinting defect). Children with AS usually undergo a considerable amount of rehabilitation during their early childhood until they achieve independent gait and are able to communicate (using varied methods of communication); however, there is currently no evidence of the effectiveness of multidisciplinary rehabilitation on the development of motor skills or on the life course of the disease. The severe developmental delay and uncoordinated movements strongly limit activities and participation in children with AS. Gross and fine motor skills generally plateau between 24 and 30 months of developmental age. Gait is clumsy, unsteady and jerky throughout the lifespan. There is a strong demand for effective motor rehabilitation from the families of children with Angelman syndrome to improve their children’s autonomy and participation [1].

Although the pathophysiology is still only partially understood, AS is related to a loss of expression or function of UBE3A, which leads to a decrease in spontaneous inhibitory transmission and an increase in spontaneous excitatory transmission: this can cause a marked excitation/inhibition imbalance that affects synaptic plasticity, particularly long-term potentiation (LTP). LTP facilitates synaptic transmission and is a mechanism of experience-dependent synaptic plasticity. Studies in AS mouse models have demonstrated that learning capacity is reversed and LTP mechanisms are restored after the reinstatement of UBE3A, supporting the hypothesis that UBE3A regulates experience-dependent learning. High-frequency stimulation has been shown to induce LTP mechanisms in animal models [2] and electrostimulation has been shown to promote motor learning in both animal models [3] and healthy volunteers [4,5]. We therefore hypothesized that neurostimulation provided simultaneously with motor rehabilitation could potentiate the effects of the motor rehabilitation.

We chose to test translingual neurostimulation (TLNS) because of the modulating properties of high frequency stimulation and the safety of cranial nerve electrostimulation in relation to epilepsy. TLNS involves high-frequency electrical stimulation of the superior surface of the tongue which is believed to alter the modulating effect of the pons, and in particular the facial and trigeminal nuclei and reticular formation which are involved in somatic motor control, notably tone, balance and postural regulation during body movements. TLNS therefore appears to be pertinent to facilitate motor learning in people with Angelman syndrome.

In this paper, we report the effects of a 4-week synergic intervention involving TLNS and goal-oriented therapy on gait ability, activities and participation in a 7-year-old child with Angelman syndrome, with a 6-month follow-up.

## 2. Detailed Case Description

### 2.1. Clinical Information

The child was diagnosed with Angelman syndrome due to a maternal deletion in the 15q11.2-13.1 chromosomal region. This is the most common genotype (68–74%) and is associated with a more severe phenotype [6]. He had severe epilepsy during infancy but had been seizure-free since the age of 5 and a half years with clonazepam and sodium valproate, with no change in dose.

The child achieved independent sitting at 14 months, crawled at 24 months, and walked independently at 5 years (only on flat, firm surfaces without obstacles). At the age of 7 years, his gross motor developmental age approximated with the Denver scale was 12 months. He had a severe limitation in adaptive behavior and severe developmental delay: he was totally dependent for all activities of daily living, such as dressing, feeding and bathing, met the criteria for autism spectrum disorders and had severe intellectual disability.

The child’s gait was jerky, with a forward lean and a ‘candlestick’ attitude of the upper limbs, and many large, involuntary arm movements. He fell often, could not stand still for more than 3 s without falling and had difficulty getting up from the floor, mainly due to axial hypotonia, hypertonia of the limbs and abnormal coordination. His daily walking distance was only around 500 m. In addition, his head-body movements were undissociated, which could put him in dangerous situations (e.g., if he turned to look at a car, he then walked onto the road).

With regard to fine motor development the child still had a palmar grasp, and was beginning to point with his index finger, corresponding to a developmental age of 7 months. Abnormal regulation of muscle tone made object handling difficult, preventing the functional use of tools and drastically reducing his engagement in meaningful activities.

In terms of language functions and behavior, the child had no speech, and he was unable to perform reliable “yes” and “no” movements with his head, thus communication was very limited. He was extremely distractible (mainly due to hyperreactivity to environmental auditory stimuli and neglect of auditory language stimuli), and displayed numerous stereotyped and repetitive behaviors, motor hyperactivity and impulsivity. In addition, he had a severe sleep disorder (which caused his parents severe fatigue and stress), which was only partially improved by melatonin and adaptations to his bed and bedroom.

### 2.2. Intervention

The intervention consisted of translingual neurostimulation (TLNS) associated with goal-oriented rehabilitation aimed to improve gait. The program lasted for 4 weeks (5 days per week) with 2 h of rehabilitation in the morning and 2 h in the afternoon. Sessions were performed in the clinic for the first two weeks. During the first hour of each session, the physiotherapist performed the rehabilitation while a parent watched, and during the second hour the parent performed the rehabilitation under the physiotherapist’s supervision. There was a 10 min break during the two parts of the session. During the second 2 weeks, the same program was performed at home by the parents with telephone support once each week. TLNS was administered simultaneously for the first hour of each session (in the clinic and at home). The child could remove the electrode if he wanted, for example to swallow his saliva.

TLNS was administered using an open-loop stimulation device developed by the Tactile Communication and Neurorehabilitation Laboratory at the University of Wisconsin in the United States. It is a voltage-controlled (19 V operating limit), capacitively coupled device that delivers rectangular positive repetitive pulses to the upper surface of the tongue by means of an irregularly shaped matrix of 143 gold-plated electrodes. The 143 electrodes are divided into nine sectors. In each sector, only one electrode is active at any instant and the other electrodes are used for current return. The system delivers triplets of 0.4 to 60 ms wide to 5 ms pulses. In each sector, the electrodes delivering the current are alternated every 20 ms (50 Hz). The biphasic pulse shape guarantees a net zero DC current, ensuring safety.

The aim of the goal-oriented rehabilitation was to improve gait ability. It included (1) gait exercises, particularly training of arm swing and direction changes, (2) balance exercises sitting on a fit ball or standing on a balance station and (3) coordination exercises such as climbing a ladder and eye-tracking exercises in sitting and while walking.

### 2.3. Tolerance of the TLNS

The child showed no signs of discomfort during or after the stimulation sessions. The only side effect was a transient increase in salivation. He accepted having the device placed in his mouth and participated well in rehabilitation while the stimulation was switched on.

### 2.4. D Gait Analysis

Gait was assessed using 3-dimensional (3D) gait analysis (Vicon system with 10 cameras) with synchronized electromyographic (EMG) analysis (Vicon system) before the intervention, at the end of the 4-week intervention and 6 months later.

The gait analysis was performed with no difficulties. Before the intervention, there was no between-limb asymmetry of kinematic of EMG variables. Velocity was low (mean 0.47 ± 0.10 m/s), step length was very short (mean 0.45 ± 0.08 m), step width was very large (mean 0.30 ± 0.05 m) reflecting poor balance, and mean percentage duration of single support was very short (30.6%). The gait was stiff-legged: foot strike occurred with the forefoot, there was no knee flexion during stance and hip flexion during swing was reduced. EMG analysis showed agonist-antagonist co-contraction of all the major muscle groups during the entire gait cycle.

At the end of the 4-week intervention, mean velocity increased to 0.78 ± 0.07 m/s, mean step length was almost doubled (0.74 ± 0.05), mean step width was reduced (0.25 ± 0.05 m) and the mean percentage duration of single support as a percentage of the gait cycle was increased (38.5 ± 3.9%). Foot strike still occurred with the forefoot; however, the knee flexed during stance and hip flexion during swing increased. Agonist-antagonist co-contractions were dramatically reduced and the pattern of muscle activation during the late stance phase and the swing phase was normal.

At 6 months, spatiotemporal, kinetic and EMG variables remained improved, although some regression had occurred, reflecting an improved gait ability in the medium term after the intervention.

### 2.5. Activity and Participation from the Family’s Perception

After 4 weeks of treatment, the parents reported a dramatic improvement in their son’s motor ability. His balance and gait were improved as well as his participation. He could walk 1 to 2 km, including on different surfaces (snow, grass and dirt paths), and arm swing was spontaneous; he was thus able to take part in family walks and was less dependent during outings. Furthermore, he showed signs of great pleasure in walking independently. With regard to activities of daily living, he could stand still for longer without falling (around 15 s), and thus could be dressed standing for example. As well as these gross motor improvements, the parents reported marked progress in other areas: the child developed clear nodding for “yes” and “no”, suddenly attributed single syllables to around 10 specific objects and people, could eat independently with a fork, and fell asleep rapidly with fewer night-time awakenings. They also noticed clear behavioral changes: he was less distractible, sought fewer intense sensory experiences, was calmer, engaged better in functional activities and could play alone with distant supervision.

Six months after ceasing the intervention and resuming his previous rehabilitation, the parents reported that the child’s gait ability and ability to eat with a fork persisted; however, the improvements in other functions disappeared over time, especially speech, sleep and behavior.

## 3. Discussion

To our knowledge, this is the first report of a rapid (4 weeks) and considerable improvement in gait ability and other voluntary and involuntary motor skills in a child with Angelman syndrome following a rehabilitation intervention.

The initial gait analysis was consistent with previous descriptions of the gait pattern [7] and the lower limb EMG characteristics [8] of children with AS in the same age group. The rapid and considerable improvement in spatiotemporal and kinetic gait variables and agonist-antagonist co-contractions, as well as the large increase in walking capacity was not expected in such a short time, suggesting that the synergic, goal-oriented intervention was effective.

A “bottom-up” modulation effect has been demonstrated in both sub-cortical and cortical areas with TLNS in patients with balance disorders and healthy volunteers [9,10,11,12,13]. The effects of the synergic intervention on gait and motor coordination suggest that the TLNS had a similar effect on the child with AS. We hypothesize that TLNS reduced neurofunctional disorders such as neuronal excitability and desynchronization and enabled the emergence of new neuronal circuits, leading to the development of new skills.

No previous studies of motor rehabilitation for AS are available, thus we cannot compare our results. However, according to our clinical experience, improvements in motor capacity usually occur very slowly in AS. We therefore suggest that the considerable, rapid improvement in motor function was due to a potentiating effect of the TLNS on the rehabilitation. This is supported by the hypothesis that the UBE3A participates in motor learning by regulating experience-dependent refinement of neuronal circuits that are dependent on sensory inputs [14] and the fact that motor learning depends mainly on synaptic plasticity mechanisms such as LTP: mechanisms which were likely positively influenced by the TLNS. It is interesting to note that the skills trained in the rehabilitation persisted at 6 months (gait ability and activities of daily living) while those which emerged without specific training regressed. Specific therapy and repetitive training for these abilities may therefore be required to consolidate motor learning. Therapy was stopped after 6 months because of difficulty accessing the TLNS device. Future studies should use standardized developmental scales to evaluate the long-term effects of the association of TLNS and goal-oriented therapy on motor development, including fine motor skills, as well as the impact on language in children with AS. The effects on electroencephalographic activity and sleep should also be determined to provide a comprehensive understanding of the effects of this therapy on AS.

## 4. Conclusions

A 4-week synergic intervention involving TLNS and goal-oriented rehabilitation resulted in a significant improvement in the functional skills and participation of a 7-year-old child with Angelman syndrome. Further studies are required to understand the mechanisms underlying this improvement, how to maintain the improvement, and to determine if such results are generalizable across all children with Angelman syndrome and other syndromes that affect synaptic plasticity.

## Data Availability

Not applicable.

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
