# Peer review of "Improvement in Gait and Participation in a Child with Angelman Syndrome after Translingual Neurostimulation Associated with Goal-Oriented Therapy: A Case Report"

_children, 2022, doi:10.3390/children9050719_

Round 1
Reviewer 1 Report
Very interesting paper.
Three considerations.
1) The case has an intellectual disability (severe? profound?): an additional diagnosis of autism spectrum disorder was made?
2) Have you performed an EEG tracing before and after the trial in order to detect any modification?
3) In order to assess other aspects besides the motor one (e.g. daily living) have you used standardized tools such as Vineland Scales?
Reviewer 2 Report
Overall, it was an interesting case study regarding the utilization of multidisciplinary TLNS for rare diseases (Angelman syndrome). The authors summarized the case concisely and effectively for the readers. Please check the attached file to address my suggestions. Once these comments are addressed, I think this paper is good for publication. Great work.

Author Response
please the attachment
